# Serum Concentration–Dose Relationship and Modulation Factors in Children and Adolescents Treated with Fluvoxamine

**DOI:** 10.3390/pharmaceutics16060772

**Published:** 2024-06-06

**Authors:** Regina Taurines, Gesa Kunkel, Stefanie Fekete, Jörg M. Fegert, Christoph Wewetzer, Christoph U. Correll, Kristian Holtkamp, Isabel Böge, Tobias Johann Renner, Hartmut Imgart, Maike Scherf-Clavel, Peter Heuschmann, Manfred Gerlach, Marcel Romanos, Karin Egberts

**Affiliations:** 1Department of Child and Adolescent Psychiatry, Psychosomatics and Psychotherapy, Centre for Mental Health, University Hospital of Wuerzburg, 97080 Wuerzburg, Germany; kunkel.gesa@googlemail.com (G.K.); fekete_s@ukw.de (S.F.); manfred.gerlach@mail.uni-wuerzburg.de (M.G.); romanos_m@ukw.de (M.R.); egberts_k@ukw.de (K.E.); 2Department of Child and Adolescent Psychiatry and Psychotherapy, University Hospital of Ulm, 89075 Ulm, Germany; joerg.fegert@uniklinik-ulm.de; 3Clinics of the City Cologne GmbH, Clinic for Child and Adolescent Psychiatry and Psychotherapy, 51109 Cologne, Germany; christoph.wewetzer@kirinus.de; 4Department of Child and Adolescent Psychiatry, Charité Universitätsmedizin Berlin, 13353 Berlin, Germany; christoph.correll@charite.de; 5Department of Psychiatry, The Zucker Hillside Hospital, Northwell Health, Glen Oaks, NY 11004, USA; 6Department of Psychiatry and Molecular Medicine, Donald and Barbara Zucker School of Medicine at Hofstra/Northwell, Hempstead, NY 11549, USA; 7German Center for Mental Health (DZPG), Partner Site Berlin, 10117 Berlin, Germany; 8DRK Fachklinik Bad Neuenahr, 53474 Bad Neuenahr-Ahrweiler, Germany; kristian.holtkamp@drk-fk-badneuenahr.de; 9Department of Child and Adolescent Psychiatry and Psychotherapeutic Medicine, Medical University of Graz, 8036 Graz, Austria; isabel.boege@medunigraz.at; 10Department of Child and Adolescent Psychiatry Ravensburg-Weissenau, ZFP South Wuerttemberg, 88427 Bad Schussenried, Germany; 11Department of Child and Adolescent Psychiatry, Psychosomatics and Psychotherapy, University Hospital of Psychiatry and Psychotherapy Tuebingen, Center of Mental Health, 72076 Tuebingen, Germany; tobias.renner@med.uni-tuebingen.de; 12German Center for Mental Health (DZPG), Partner Site Tuebingen, 72076 Tuebingen, Germany; 13Parkland-Clinic, Clinic for Psychosomatics and Psychotherapy, Academic Teaching Hospital for the University Gießen, 34537 Bad Wildungen, Germany; hartmut.imgart@parkland-klinik.de; 14Department of Psychiatry, Psychosomatics and Psychotherapy, Center of Mental Health, University Hospital Wuerzburg, 97080 Wuerzburg, Germany; scherf_m@ukw.de; 15Clinical Trial Center Wuerzburg, University Hospital Wuerzburg, 97080 Wuerzburg, Germany; peter.heuschmann@uni-wuerzburg.de; 16Institute of Clinical Epidemiology and Biometry, University of Wuerzburg, 97080 Wuerzburg, Germany

**Keywords:** fluvoxamine, children, serum concentration, obsessive compulsive disorder, therapeutic drug monitoring, pharmacovigilance

## Abstract

Introduction: Fluvoxamine is used in children and adolescents (‘youths’) for treating obsessive compulsive disorder (OCD) but also off-label for depressive and anxiety disorders. This study aimed to investigate the relationship between fluvoxamine dose and serum concentrations, independent correlates of fluvoxamine concentrations, and a preliminary therapeutic reference range (TRR) for youths with OCD and treatment response. Methods: Multicenter naturalistic data of a therapeutic drug monitoring service, as well as prospective data of the ‘TDM Vigil study’ (EudraCT 2013-004881-33), were analyzed. Patient and treatment characteristics were assessed by standardized measures, including Clinical Global Impressions—Severity (CGI-S) and —Change (CGI-I), with CGI-I of much or very much improved defining treatment response and adverse drug reactions using the Udvalg for Kliniske Undersogelser (UKU) Side Effect Rating Scale. Multivariable regression analysis was used to evaluate the influence of sex, age, body weight, body mass index (BMI), and fluvoxamine dose on fluvoxamine serum concentrations. Results: The study included 70 youths (age = 6.7–19.6 years, OCD = 78%, mean fluvoxamine dose = 140.4 (range = 25–300) mg/d). A weak positive correlation between daily dose and steady-state trough serum concentrations was found (r_s_ = 0.34, *p* = 0.004), with dose variation explaining 16.2% of serum concentration variability. Multivariable correlates explaining 25.3% of the variance of fluvoxamine concentrations included higher fluvoxamine dose and lower BMI. Considering responders with OCD, the estimated TRR for youths was 55–371 ng/mL, exceeding the TRR for adults with depression of 60–230 ng/mL. Discussion: These preliminary data contribute to the definition of a TRR in youth with OCD treated with fluvoxamine and identify higher BMI as a moderator of lower fluvoxamine concentrations.

## 1. Introduction

The selective serotonin reuptake inhibitor fluvoxamine is approved by the Food and Drug Administration as well as the European Medicines Agency to treat obsessive compulsive disorder (OCD) in children and adolescents aged ≥ 8 years. However, in clinical practice, fluvoxamine is also used off-label in patients younger than 18 years for other conditions, such as depressive episodes or anxiety disorders. Randomized controlled trials (RCTs) have confirmed the effectiveness/efficacy of fluvoxamine in children and adolescents (‘youths’) treating OCD, with the most common side effects of insomnia and asthenia [1] and recommended daily doses up to 200 mg/d (max. 300 mg/d for adolescents aged 12–17 years, U.S. Food and Drug Administration (FDA)). Fluvoxamine also significantly reduced symptoms of anxiety and increased response rates in a U.S. RCT involving 128 pediatric patients with social phobia, separation anxiety disorder, or general anxiety disorder [2].

Fluvoxamine is metabolized by the cytochrome P450 (CYP) enzymes CYP1A2 and CYP2D6 to inactive metabolites. Fluvoxamine is also an inhibitor of several CYP enzymes (strong: CYP1A2, CYP2C19; moderate: CYP2C9 and CYP3A4; weak: CYP2D6 [3,4,5,6]).

Therapeutic drug monitoring (TDM) represents a valid method to address pharmacokinetic variability by individual dose adjustment based on measured serum concentrations [7]. In neuropsychopharmacology, TDM is generally recommended for children and adolescents due to developmental peculiarities in pharmacokinetics and pharmacodynamics and the frequent off-label use of psychotropic medications with less knowledge about effective dose ranges [8]. Regarding SSRIs, TDM is especially helpful in children and adolescents to shed light on individual variability of pharmacokinetics, and TDM can help explain non-response or partial response, understand drug–drug interactions, detect adherence problems, and investigate the impact of variations in pharmacokinetic gene effects [9]. For adult patients with OCD, no fluvoxamine concentration recommendations exist, but for adults with depression, serum fluvoxamine concentrations in the range of 60–230 ng/mL are recommended as the therapeutic reference range (TRR), with a recommendation level of 2 [8]. In children and adolescents, however, the relationships between fluvoxamine dose and blood concentrations have not been clarified yet, and a TRR is not defined for different pediatric age groups or indications.

Reviewing the few existing pharmacokinetic and TDM studies in pediatric patients, in a sample of 34 young patients [10], nonlinear pharmacokinetics over the dose range studied (up to 150 mg/d) and a higher exposure to fluvoxamine was reported in children than adolescents, whereas adolescents and adults appeared to have similar exposure to fluvoxamine. Biener and colleagues found in a doctoral thesis in 54 young patients a correlation between fluvoxamine dose and serum concentration [11,12]. In a retrospective chart review study on fluvoxamine–clomipramine combination therapy in six minors with OCD, dose-dependent inhibition of clomipramine metabolism by fluvoxamine was observed with no serious adverse events due to the co-treatment with fluvoxamine [13].

Besides these few studies, there are no clinical studies on fluvoxamine serum levels and TDM in minors with psychiatric disorders. Therefore, the primary aim of this study was to assess the relationship between daily dose and serum concentration in children and adolescents treated with fluvoxamine, using multicenter data from a routine TDM service and of the ‘TDM Vigil’ pharmacovigilance study (EUDRA CT 2013-004881-33). As a secondary aim, the influence of several patient and treatment characteristics as possible moderating or mediating factors of fluvoxamine serum concentrations was investigated. Furthermore, it was evaluated whether the recommended range for blood concentrations in adults might be valid and applicable to children and adolescents or whether a different TRR for children and adolescents can be identified.

## 2. Subjects and Methods

### 2.1. Setting and Study Population

All patients who were administered fluvoxamine and for whom a concentration determination was carried out were eligible for the study, regardless of the diagnosis and treatment setting (inpatient, outpatient, day clinic). Patient and prescription data, as well as blood samples, were collected from four university hospital departments (Berlin, Tuebingen, Ulm, and Wuerzburg) and four child and adolescent psychiatric state hospitals (Bad Neuenahr, Bad Wildungen, Cologne Holweide, and Ravensburg) in Germany and Austria between 2006 and 2019 (Table 1). All participating hospitals are members of the competence network for TDM in child and adolescent psychiatry (www.tdm-kjp.com, accessed on 1 April 2024 [14]). The majority (86%, *n* = 60) of the patients were recruited within a clinical routine TDM service, and 14% of the patients (*n* = 10) were enrolled via the TDM Vigil study, which is described in detail elsewhere [15]. All patients received a physical–neurological and psychiatric examination, assessment of vital signs, height, body weight, and laboratory analyses for hepatic and renal function. Patient characteristics (sex, age, diagnosis according to ICD-10, comorbidity, nicotine use), data of drug treatment (e.g., daily dose of fluvoxamine, type and dose of any psychiatric co-medications), and outcome data for global illness severity and improvement as well as adverse drug reactions (ADRs) were collected in a standardized way (see below). Furthermore, the date, reason for TDM analysis (e.g., ‘dose adjustment’ or ‘compliance control’), and the symptoms intended to treat with the medication (e.g., ‘obsessive compulsive behavior’ or ‘depressive symptoms’) were documented. Patients were excluded from the study if steady-state conditions for blood taking were not fulfilled (i.e., last fluvoxamine dose adjustment < 5 days ago), relevant data were missing (e.g., daily dose, relevant patient information), if there was an absolute contraindication for fluvoxamine, or if the patients participated in a clinical trial other than TDM Vigil.

The study was approved by the local ethics committee of the University of Wuerzburg (study numbers 27/04, 301/13) and carried out according to the Declaration of Helsinki. In the subgroup in which the investigation of serum concentrations was part of the routine clinical blood tests, written consent to analyze the anonymized data was not necessary. Within the prospective TDM Vigil study, written informed consent was obtained from adults aged 18–19 and legal guardians of minors, with written informed assent by minors from the age of 14 years (for details, see [15]).

### 2.2. Measurement of Fluvoxamine Serum Concentrations

All analyses of fluvoxamine serum concentrations were performed by the specialized TDM laboratory of the Center of Mental Health of the University Hospital Wuerzburg according to the consensus guidelines of TDM in Psychiatry of the German Society for Neuropsychopharmacology and Pharmacopsychiatry (Arbeitsgemeinschaft für Neuropsychopharmakologie und Pharmakopsychiatrie, AGNP [8]). In steady-state conditions (i.e., after ≥5 days after the last fluvoxamine dose adjustment), blood withdrawal from cubital veins was performed in 7.5 mL monovettes without anticoagulants and additives as trough value before the first daily intake of fluvoxamine. The elimination half-life of fluvoxamine is 15–20 h. Steady-state plasma fluvoxamine concentrations are reached after at least 5 days after initiation of therapy [16]. The date and time of blood withdrawal were noted. The blood was centrifuged at 1800 g for 10 min and analyzed immediately (samples from Wuerzburg) or within a few days after mailing according to standard procedures to the central TDM laboratory.

Fluvoxamine serum concentrations were analyzed by an automated column-switching method coupled to an isocratic high-performance liquid chromatography (HPLC) system and a variable ultraviolet detector (for the method, see [17]). The intra-assay coefficients of variation determined from 10 analyses (73 and 293 ng/mL) were, in general, <1%. The inter-assay variability was, in general, <1.5%. The method was linear in a range of 10–730 ng/mL (r^2^ = 0.99), and the lower limit of quantification was 10 ng/mL. Samples above a concentration of 730 ng/mL were diluted with isotonic saline solution, and different dilutions of the sample were measured and compared with the undiluted sample. Chemicals and solvents with level of purity and fluvoxamine for calibration and controls were purchased commercially from Sigma-Aldrich, Munich, Germany. For patients with more than one concentration determination, the chronologically last available measurement was selected for this study.

### 2.3. Assessment of Therapeutic Outcomes

To assess the severity of psychopathology and the change in symptomatology at the time of blood withdrawal, the Clinical Global Impressions Scale was used (severity: CGI-S; improvement: CGI-I), and the change (improvement) of global illness severity (CGI-I) was applied as a measure for effectiveness [18]. The following categories were used for CGI-I according to the CGI manual: 1 = very much improved, 2 = much improved, 3 = minimally improved, 4 = unchanged, 5 = minimally worse, and 6 = much worse. Patients with a CGI-I score of 1 and 2 were defined as ‘responding to treatment’, with CGI scores of 3, 4, 5, or 6 as ‘non-responders’.

The nature and severity of ADRs at the time period before blood taking were assessed using the Udvalg for Kliniske Undersogelser Side Effect Rating Scale [19] with the following categorization: 0 = no side effects, 1 = mild, 2 = moderate, and 3 = severe side effects.

### 2.4. Data Analysis

Statistical analyses were performed with the software SPSS, version 26. Means and standard deviations, medians, and interquartile ranges (IQRs) were calculated for descriptive analyses. Kurtosis and skewness tests were used to evaluate variables for Gaussian distribution. The Spearman rank correlation coefficient (r_s_) was applied for non-Gaussian-distributed variables and the Pearson coefficient (r_p_) for Gaussian-distributed variables. Group differences were analyzed by independent two-tailed *t*-test or Mann–Whitney U test for normally distributed and non-normally distributed variables, respectively. Multivariable linear regression analysis was used to determine possible moderating factors on fluvoxamine concentrations, evaluating age, sex, body weight, body mass index (BMI), and fluvoxamine dose as independent variables. In the absence of valid data on a therapeutic reference range in pediatric patients, a method proposed in the literature was used to estimate a TRR based on the IQR of drug concentrations (25th–75th percent range) in the blood of patients responding to drug therapy [20,21,22]. Statistical significance was defined as *p* ≤ 0.05. All values are presented as mean ± SD or as median and IQR wherever appropriate.

## 3. Results

### 3.1. Study Population

The study population comprised 70 patients (54.3% female) with a mean (SD, median) age of 14.8 (2.4, 15) years, of whom 20 (28.6%) were younger than 14 years (Table 1). The majority (77.9%) had a diagnosis of OCD (ICD-10 F42), 11.8% presented with eating disorders (ICD-10 F50), 8.8% with a depressive episode (ICD-10 F32), and 8.8% with hyperkinetic disorders (ICD-10 F90). The severity of symptomatology was in most patients classified as ‘markedly ill’ (40.4%), followed by ‘severely ill’ (19.3%) or ‘moderately ill’ (17.5%).

The majority of patients (64.1%) for whom detailed information on the dosing regimen was available received fluvoxamine twice daily. Altogether, 34.8% of all patients received one or more concomitant psychotropic medications. Fluvoxamine monotherapy was significantly more frequent in females 79.4% (*n* = 27) than in males 50% (*n* = 16) (X^2^ = 6281; df = 1; *p* = 0.012) and children 84.2% (*n* = 19) compared to adolescents 57.4% (*n* = 27) (X^2^ = 4269; df = 1; *p* = 0.039).

### 3.2. Fluvoxamine Dose in Relation to Other Covariates

Patients of the whole transdiagnostic sample were treated with an average of 140.4 mg/d (SD 61.4; range 25–300 mg/d) fluvoxamine (Table 2). The group of all patients with OCD (*n* = 53) were treated with significantly higher daily doses (149.5 ± 62.7 mg/d) compared to all patients (*n* = 15) with other diagnoses (113.3 ± 50.8 mg/d) (t = −2.050; *p* = 0.044); data on the whole group of patients with OCD and the whole group of patients with other diagnoses are not additionally shown in Table 2. The daily dose did not differ in subgroups classified according to age (*p* = 0.239), sex (*p* = 0.605), or mode of pharmacotherapy (monotherapy/co-medication) (*p* = 0.087). The mean body weight-adjusted dose was 2.8 mg/kg (SD 1.3, range 0.7–5.8 mg/kg), and there was no statistically significant difference between boys and girls (*p* = 0.63) nor between children and adolescents (*p* = 0.17).

### 3.3. Fluvoxamine Concentration in Relation to Fluvoxamine Dose and Other Covariates

The mean (SD) fluvoxamine concentration (*n* = 70) was 186.0 ng/mL (159.4). A large inter-patient variability of fluvoxamine concentrations was shown (range 12.0–754.0 ng/mg, IQR 76.5–243.5). The mean dose-corrected fluvoxamine concentration (C/D) was 1.4 (ng/mL)/(mg/day) (SD 1.3, range 0.08–7.7). Table 2 shows the measured serum concentrations in the total population and different subsamples. In the total sample, a positive, weak correlation between daily doses and fluvoxamine concentrations was found (r_s_ = 0.34, *p* = 0.004), with the variation in dose explaining 16.2% of the variability in serum concentrations (r_s_^2^ = 0.16) (Figure 1). When using body weight-adjusted doses, similar correlation results were calculated (r_s_ = 0.47; r_s_^2^ = 0.22; *p* < 0.001). A positive correlation of daily dose and serum concentrations was calculated for most of the subgroups (see Table 2).

Mean fluvoxamine serum concentration did not significantly differ in the following subgroups: mono-/polypharmacy (*p* = 0.45); ADRs yes/no (*p* = 0.90); responder/non-responder (*p* = 0.50); OCD/other diagnosis (*p* = 0.24).

Considering potential interaction effects with psychotropic medication, we identified four patients on CYP2D6 inhibiting co-medication (fluoxetine, sertraline, and clomipramine). Two patients on fluvoxamine plus fluoxetine (strong inhibitor) showed higher dose-corrected serum concentrations of fluvoxamine than expected (assessed according to [23]). Serum concentrations in the two other patients on sertraline and clomipramine (weak inhibitors) showed no alterations (assessed according to [23]).

Multiple linear regression analysis revealed a significant effect of the covariate BMI (*p* = 0.03), with higher BMI being associated with lower (dose-corrected) fluvoxamine concentrations (Figure 2). Together, BMI and fluvoxamine dose explained 25.3% of the variance in concentrations. Age (*p* = 0.16) and sex (*p* = 0.76) were not significantly associated with fluvoxamine serum concentrations. An influence of cigarette smoking could be expected (by induction of CYP1A2) but was not analyzable, as only one smoker was documented in the sample. Also, the effect of co-medications could not be determined via multiple linear regression analysis, as the number of patients with a clinically relevant potential of interaction with the metabolism of fluvoxamine was too small (*n* = 4, see above).

### 3.4. Clinical Positive and Negative Effects of Fluvoxamine Treatment

The CGI-I scale was rated in 50 patients, whereby the clinical outcome was judged as not assessable in 6 patients. Out of a total of 44 patients with an assessable score, 20.5% (*n* = 9) were rated as ‘very much improved’ and 31.8% (*n* = 14) as ‘much improved’ and together were classified as responders (52.3%, *n* = 23). In contrast, 47.7% (*n* = 21) of the patients were classified as non-responders as their state was rated as ‘minimally improved’ (31.8%), ‘unchanged’ (13.6%) or ‘minimally worse’ (2.3%) (none as ‘much worse’ or ‘very much worse’). There was no significant difference in fluvoxamine concentrations in responders compared to non-responders (*p* = 0.50) (Table 2). No statistically significant correlation of serum concentrations with CGI-I scores was observed (*p* = 0.056).

ADRs (one or more) were documented in 37.5% (*n* = 21) of the patients. In the group of patients with ADRs and a documented rating on the severity of the ADRs (*n* = 15), 12 patients (80%) were rated as ‘mild’, and 3 patients (20%) as ‘moderate’. ‘Severe’ ADRs were not documented in any of the patients. There was no significant correlation between fluvoxamine serum concentrations and the severity of ADRs (*p* = 0.928). In 29 patients, details on the type of ADRs were documented. With 48.3% (*n* = 14), sedation/drowsiness was the most frequently reported ADR. Other ADRs were rare and comprised cardiovascular ADRs (*n* = 3), extrapyramidal symptoms, EPSs (*n* = 2), feeling of inner tension/agitation (*n* = 2), skin irritation (*n* = 1), and hyposalivation (*n* = 1). Fluvoxamine concentrations did not differ significantly in patients with and without any kind of ADRs (*p* = 0.899; U = 360,000).

In the above-mentioned two patients with the CYP2D6 inhibiting co-medication fluoxetine and with concentrations of 333 ng/mL and 155 ng/mL (both showed higher dose-corrected fluvoxamine serum concentrations), no side effects were documented.

In three patients (4%), fluvoxamine serum concentration reached the so-called laboratory alert level (≥500 ng/mL), a threshold that obliges the laboratory to provide feedback immediately to the prescribing physician. In one of these three patients, no side effects appeared; in the second one, EPSs were documented; data on ADRs were unfortunately missing in the third patient.

### 3.5. Estimation of a Preliminary Therapeutic Reference Range of the Fluvoxamine Concentration in Children and Adolescents

The comparison of fluvoxamine concentrations in the present pediatric sample with the currently recommended TRR for adult patients with major depressive disorder (60–230 ng/mL) [8] revealed that 51.4% (*n* = 36) of all measured fluvoxamine concentrations were within the TRR for adults, 17.1% (*n* = 12) were below and 31.4% (*n* = 22) above the TRR for adults.

Derived from the concentrations of all patients who responded to drug therapy (*n* = 23, mean fluvoxamine dose: 139.1 ± 69.4 mg/d), a preliminary transdiagnostic TRR of fluvoxamine concentration for children and adolescents was determined, leading to a preliminary suggested therapeutic range with a lower limit of 38 ng/mL and an upper limit of 239 ng/mL. Considering responders in youths with the diagnosis OCD (*n* = 18; mean fluvoxamine dose: 143.1 ± 74.7 mg/d, mean serum concentrations 193.3 ± 202.2 ng/mL), the calculated diagnosis-specific therapeutic reference range would be higher with 55–371 ng/mL exceeding the TRR defined for adults with depression (60–230 ng/mL [8]).

## 4. Discussion

Data on the pharmacokinetics of fluvoxamine in children and adolescents are scarce. In this, to our knowledge, the largest TDM sample of children and adolescents treated with fluvoxamine studied to date, a positive, albeit weak, linear relationship between daily fluvoxamine dose and concentration and high interindividual variability of serum concentrations was found. The results pertaining to fluvoxamine treatment responders further suggest a similar upper limit but a lower lower limit of a possible transdiagnostic TRR for youths compared to adults. Considering only patients with OCD, the calculated upper limit of TRR was higher in youths than the TRR currently defined for adults with depression (see below for details).

In the present study, similar average daily doses of fluvoxamine (140 mg/d) were administered as in the previously largest TDM sample of 54 minors with mainly OCD diagnoses in a doctoral thesis (150 mg/d) [11,12]. A weak linear relationship between daily fluvoxamine dose and serum concentration was found, which is also in line with the results of Biener and coworkers [11,12]. In the small subsample of 20 children aged < 14 years, a strong correlation was found in our study, possibly due to the significantly higher proportion of children on monotherapy compared to adolescents. Labellarte and colleagues [10], however, reported nonlinearity in a smaller sample of 16 children and 18 adolescents treated with fluvoxamine. Studies in adult patients showed conflicting results, too [24,25,26,27,28]. The result of nonlinearity in smaller studies might be modulated by CYP2D6 gene variants, which were not considered in any study and can remarkably affect plasma exposure to fluvoxamine [29,30,31,32]. Unfortunately, no data on metabolizer status were available in the present sample.

In the current study, a wide distribution of serum concentrations was found, and no significant impact of sex and age, a result in line with Biener et al.’s TDM data in minors [11,12]. Labellarte and coworkers, however, reported an effect of these covariates. After normalization for body weight, they found higher fluvoxamine plasma concentrations and lower oral clearance in children compared to adolescents [10]. In the small group of 16 minors, they also found higher mean peak plasma concentrations in females compared to males. In adults, no effect of sex was reported [25]. In the present sample, BMI and dose explained 25.3% of the variance in concentrations; dose alone only 16.2%. A higher BMI resulted in relatively lower serum concentrations, implying that patients with a high BMI might need a higher dose. Neither in studies in minors [10,11,12] nor in adults was such a moderating effect of BMI on fluvoxamine serum concentration reported [16,33]. Fluvoxamine presents with a serum protein binding of about 80% [3]. BMI might impact the volume of distribution (as reported, e.g., for escitalopram [34]), or it might alter protein binding capacity via changes in the serum proteome [35]. Given inconsistent results in the literature regarding the relationship between BMI and fluvoxamine concentrations in youths and adults, further studies in this field are needed to clarify, whether incorporating BMI can better guide therapeutic decision making with respect to the dosing strategy of fluvoxamine in youths. In some clinical samples, an effect of cigarette smoking on steady-state fluvoxamine concentrations via induction of CYP1A2 was suggested, with lower concentration/dose ratios [36,37], however, in others [38] no major impact could be described. As there was only one smoker in the present sample, such analyses were not possible.

In nearly half of the patients, a suitable therapeutic response to fluvoxamine treatment was reported. Findings on a relationship between serum concentrations and response to fluvoxamine in adults and minors are heterogeneous [10,11,12,25,26,27,38,39,40]; in the present sample, no such correlation was observed. ADRs were documented in more than a third of the children and adolescents, a result that is consistent with the work of Biener et al. in minors (35% [11,12]) as well as with studies in adults (42–50% [41,42]). No serious adverse effects were documented in any of the patients of this study monitored by TDM. In accordance with the present results, no clear correlation between fluvoxamine serum concentrations and the occurrence or severity of adverse events has been reported in the literature [26,28].

TDM of patients taking fluvoxamine is recommended for personalized pharmacotherapy, both for dose titration and special indications, such as the assessment of medication adherence. For adult patients with depression, the TRR has been defined as 60–230 ng/mL based on studies with (flexible) fluvoxamine doses between 200 and 300 mg/d [25], 150 and 200 mg [26], and 25 and 200 mg/d [8,43]. A comparison of the present data with the TRR for adults with depression revealed that about half (51.4%) of all measured fluvoxamine concentrations were within the TRR for adults with depression, and almost one-third (31.4%) were above the TRR for adults. For OCD in adults, no TRR has been defined so far, as there are insufficient data on the topic [44]. In children and adolescents treated with fluvoxamine, a TRR has not yet been defined for any indication and no controlled TDM studies with fixed dose designs nor positron emission tomography (PET) studies to measure fluvoxamine concentration-serotonin transporter occupancy relationships are available. Considering the IQR of drug concentrations (25th–75th percent range) in the blood of all responders to the drug therapy [20,21], data of this study suggest a rather identical upper fluvoxamine concentration limit in minors (240 ng/mL) and adults (230 ng/mL), and a slightly lower lower limit of about 40 ng/mL (adults 60 ng/mL) as a preliminary transdiagnostic TRR for youths compared to thresholds in adults with major depressive disorder. Considering only responders with an OCD diagnosis, the calculated TRR would be higher with 55–371 ng/mL, exceeding the upper limit of the TRR in adults with depression. This result fits with the finding that a higher mean multiplication factor was calculated for fluvoxamine in both children aged 6 to 11 years (1.1) and adolescents aged 12–18 years (0.41) to determine the dose-related concentration than in adults (0.23) [23]. The pediatric patients in our sample were, on average, not treated with higher fluvoxamine doses than the doses on which the TRR for adult patients with depression is based [8]. In PET studies of adult patients, occupancy of serotonin transporters in the brain correlates well with fluvoxamine blood concentrations, while relatively small clinical doses of 50 mg fluvoxamine maleate consistently occupied approximately 80% of serotonin transporters [45]. As long as there are no PET studies in youths that show differing results, it has to be carefully considered to exceed the recommended upper limit of TRR of adults treating children with fluvoxamine. Nevertheless, as it is known that adult patients with OCD often require higher SSRI doses than patients with major depressive disorder [46], it is possible that also in children, higher SSRI concentrations are needed to lead to a response in OCD symptoms (see, e.g., [47]). This hypothesis requires further testing in future TDM studies in children and adolescents, and the TRR for OCD in adults should also be examined.

The findings of the present study must be interpreted in the context of several limitations. First, although the present sample of 70 children and adolescents is the biggest reported to date, it is still limited, and the subgroups of subjects (boys/girls, children/adolescents, monotherapy/co-medication, OCD/non-OCD, responders/non-responders, etc.) are too small to identify variables with small effect sizes. Second, the present data go along with the typical limitations of a naturalistic–observational study, including heterogeneous patient groups and non-standardization of dose regimes and length of treatment before TDM analysis. Third, treatment response was assessed using a global illness measurement, the CGI-I, and not with disease-specific, more detailed assessments of psychopathology. Fourth, this naturalistic study design is not well suited for determining concentration–effect relationships [21], as placebo-responders and patients with ADRs, who are likely to receive lower doses, were not excluded from the assessment, nor were non-responders who likely receive higher doses. Fifth, there was no rigorous control of medication adherence. Finally, the genotypes of CYP2D6 were not investigated, which could also be a reason for the pharmacokinetic variability shown. In the future, genotyping of CYP2D6 in addition to TDM might help to better contextualize the measured serum concentrations and finally achieve a more personalized pharmacotherapy. Based on these limitations, to define a TRR, further studies with a more controlled design are required, with larger sample sizes and fixed dosing regimens, fixed time points of clinical evaluation, and use of more specific psychometric instruments to capture treatment response in patients with different diagnoses or to investigate dose-dependent ADRs.

As a strength of the present study, naturalistic TDM approaches allow the collection of data on dose–concentration relationships in ‘real world patients’, who are defined by diverse individual characteristics and concomitant medications. In RCTs, patients who use multiple psychotropic drugs are usually excluded. About one-third of the young patients in the present sample received at least one concomitant psychotropic drug in addition to fluvoxamine, mainly antipsychotics (34.8%). Only 6% were taking antidepressants with potential CYP2D6-inhibiting effects, so these numbers were too small to influence the findings substantially and draw valid conclusions from the present data.

## 5. Conclusions

In the present study, TDM of youths treated with fluvoxamine was studied in the up-to-date largest pediatric sample in a real-life setting, thereby significantly increasing the amount of available data in this vulnerable population. A weak correlation between daily dose and fluvoxamine concentration was found, as well as indications that BMI should be taken into account when titrating the dose, with higher BMI possibly requiring higher fluvoxamine doses to achieve similar fluvoxamine concentrations as in youths with lower BMI. Considering youth with an OCD diagnosis who were rated as treatment responders to fluvoxamine, the calculated TRR would exceed the TRR defined for adults with depression. Due to the high interindividual variability of serum concentrations, TDM provides an effective pharmacovigilance tool acknowledging individual pharmacokinetic parameters in the pediatric population that should be used more often to inform clinical care.

## Figures and Tables

**Figure 1 pharmaceutics-16-00772-f001:**
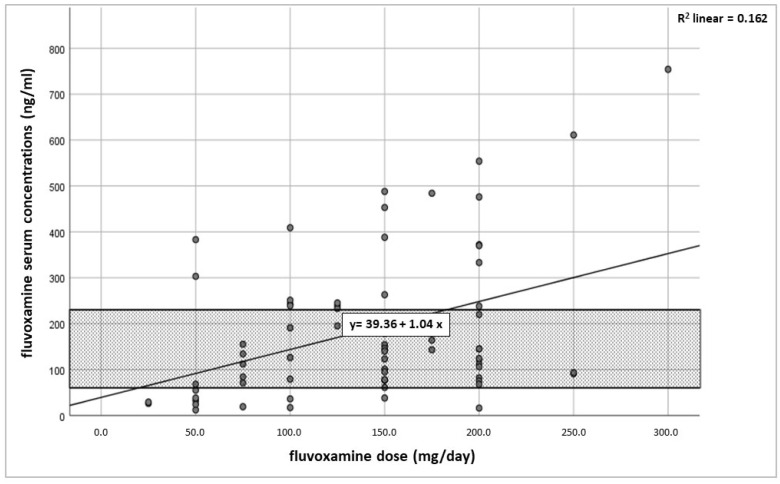
Relationship between fluvoxamine dose per day and the fluvoxamine steady-state trough serum concentrations for *n* = 70 patients. The recommended therapeutic range of fluvoxamine concentrations in adults (60–230 ng/mL) is highlighted.

**Figure 2 pharmaceutics-16-00772-f002:**
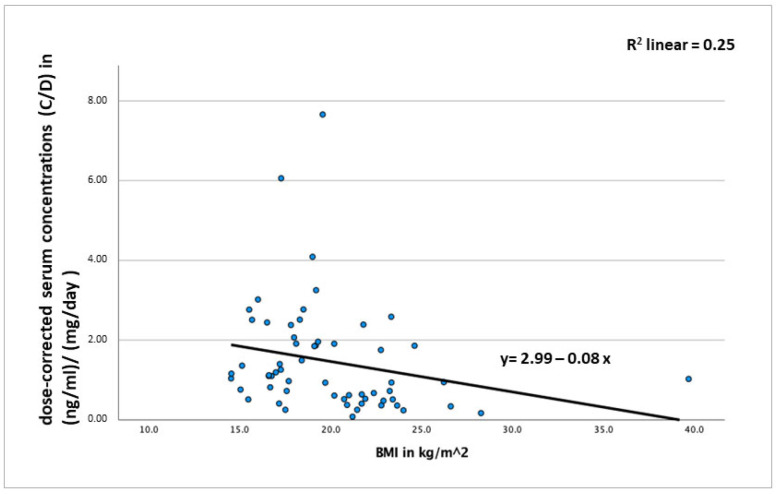
Correlation between BMI (kg/m^2^) and dose-corrected serum concentrations of fluvoxamine (C/D) in (ng/mL)/(mg/day) in *n* = 60 patients.

**Table 1 pharmaceutics-16-00772-t001:** Characteristics of study population (N = 70).

**Clinical Center, N (%)**			
Ulm		25	(35.7)
Wuerzburg		24	(34.3)
Cologne		10	(14.3)
Berlin		4	(5.7)
Bad Neuenahr		3	(4.3)
Ravensburg		2	(2.9)
Tuebingen		1	(1.4)
Bad Wildungen		1	(1.4)
**Sex,** N (%)			
Female		38	(54.3)
Male		32	(45.7)
**Age** (years), mean ± SD, median		14.8 ± 2.4, 15	
**Children** < 14 years, N (%)		20	(28.6)
**Adolescents** ≥ 14 years, N (%)		50	(71.4)
**Weight** (kg), N = 60, mean ± SD, median		52.0 ± 16.9, 50.5	
**Height** (cm), N = 61, mean ± SD, median		159.6 ± 13.3, 162.0	
**BMI** (kg/m^2^), N = 60, mean ± SD, median		20.0 ± 4.1, 19.2	
**Smoking**, N = 57, N (%)		1	(1.43)
**Most common ICD diagnosis**, N = 68, N (%), multiple entries			
F 42	Obsessive compulsive disorder	53	(77.9)
F 50	Eating disorders	8	(11.8)
F 32	Depressive episode	6	(8.8)
F 90	Hyperkinetic disorders	6	(8.8)
**Severity of illness (CGI-S)**, N = 57, N (%)			
Not assessable		3	(5.3)
Not at all ill		1	(1.8)
Mildly ill		6	(10.5)
Moderately ill		10	(17.5)
Markedly ill		23	(40.4)
Severely ill		11	(19.3)
Extremely ill		3	(5.3)
**Fluvoxamine monotherapy**, N = 66, N (%)		43	(65.2)
**Psychiatric co-medication**, N = 66, N (%), multiple entries			
Antipsychotics		23	(34.8)
(aripiprazole, haloperidol, olanzapine, melperone, pipamperone, chlorprothixene, quetiapine, and risperidone)			
Tranquilizer		1	(1.5)
Antidepressants		4	(6.1)
(clomipramine, fluoxetine, and sertraline)			
Stimulants		1	(1.5)
**Clinical outcome (CGI-I)**, total N = 50, N = 44 with assessable score, N (%)			
(Not assessable 6)			
Very much improved ^(1^*^)^		9	(20.5)
Much improved ^(2^*^)^		14	(31.8)
Minimally improved ^(3^*^)^		14	(31.8)
Unchanged ^(4^*^)^		6	(13.6)
Minimally worse ^(5^*^)^		1	(2.3)
Much worse ^(6^*^)^			-
‘Responder’ ^(1+2^*^)^		23	(52.3)
‘Non-responder’ ^(3+4+5+6^*^)^		21	(47.7)
**Adverse Drug reactions (UKU)**, N = 56, N (%)			
Number of patients with ADRs		21	(37.5)
**Severity of ADRs (UKU)**, N = 15, N (%)			
Mild		12	(80.0)
Moderate		3	(20.0)
Severe		0	(0.0)

CGI-I = Clinical Global Impression Scale—Improvement, * = categories according to the CGI-I, CGI-S = Clinical Global Impression Scale—Severity, UKU = Udvalg for Kliniske Undersogelser Side Effect Rating Scale, N = number of patients (The N-numbers can deviate from the total number of patients, as complete data were not available for every patient), SD = standard deviation.

**Table 2 pharmaceutics-16-00772-t002:** Fluvoxamine daily doses and serum concentrations in different subsamples.

**Patients (N)**	**Fluvoxamine Daily Dose** **Mean ± SD** **Median** **(Range) (mg/day)**	**Fluvoxamine Concentration** **Mean ± SD** **Median** **(IQR) (ng/mL)**	**Correlation Daily Dose–Fluvoxamine Concentration** **(r_s,_ *p*)**	**Group Differences in Fluvoxamine Concentrations** **(*p*)**
All (70)	140.4 ± 61.4 150 (25–300)	186.0 ± 159.4 137 (76.5–243.5)	0.34 <0.004	
Children (20)	125 ± 60.2 150 (25–200)	169.4 ± 169.8 83.0 (56.5–248.0)	0.75 <0.001	0.23
Adolescents (50)	146.5 ± 61.4 150 (25–300)	192.6 ± 156.4 145.0 (88.0–243.5)	0.13 0.381	
Boys (32)	144.5 ± 69.5 150 (25–300)	193.0 ± 186.6 135.5 (78.3–243.3)	0.63 <0.001	0.91
Girls (38)	136.8 ± 54.4 150 (25–200)	180.0 ± 134.6 137.0 (73.3–243.5)	0.03 0.862	
Monotherapy (43)	130.2 ± 64.0 150 (25–300)	171.0 ± 157.2 123 (68.0–239.0)	0.26 0.093	0.45
Co-medication (23)	157.6 ± 54.6 150 (50–250)	205.6 ± 172.9 140 (78.0–333.0)	0.43 0.041	
Responders transdiagnostic (23)	139.1± 69.4 150 (25–300)	171.4 ± 186.0 84 (38.0–239.0)	0.39 <0.066	0.50
Non-Responders transdiagnostic (21)	153.6 ± 57.6. 150 (50–250)	214.4 ± 158.6 145 (123.0–257.0)	0.06 0.29	
Responders OCD (18)	143.1 ± 74.7 150 (25–300)	193.3 ± 202.3 (55.3–370.5)	0.39 0.111	0.41
Non-Responders OCD (17)	166.2 ± 53.0 175 (75–250)	226.5 ± 167.4 145.0 (122.5–324.0)	0.01 0.65	

OCD = obsessive compulsive disorder.

## Data Availability

The original contributions presented in the study are included in the article; further inquiries can be directed to the corresponding author.

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
