# Peer review of "Serum Concentration–Dose Relationship and Modulation Factors in Children and Adolescents Treated with Fluvoxamine"

_pharmaceutics, 2024, doi:10.3390/pharmaceutics16060772_

Round 1

Reviewer 1 Report

Comments and Suggestions for Authors

In this manuscript by Taurines R and colleagues, the authors have analysed the relationship between serum concentrations and the dose of fluvoxamine administered to children and adolescents for treating obsessive-compulsive-disorders (OCD). Based on their multiple linear regression analysis, the authors have identified the high body mass index (BMI) as main source of variability among patients being associated with lower (dose-corrected) fluvoxamine serum concentrations. Finally, the authors have proposed a therapeutic range based on fluvoxamine serum concetrations measured in OCD responders.

In my opinion this analysis has been well conducted and conclusions seem to be adequately supported by results. However, there are some concerns that should be addressed:

1) 2.2 Measurement of fluvoxamine serum concentrations. The authors should provide more datails on the bioanlytical method used. Was this method already published or validated elsewhere?

2) 2.4 Data analysis. The dose used for the multiple linear regression was normalized on body weight? Moreover, the serum fluvoxamine concentration was dose-corrected?

3) 3.3 Fluvoxamine concentration in relation to fluvoxamine dose and other covariates. The odd ratio or the parameter estimate should be added for the BMI covariate.

4) Line 305. "Significant moderating effect" should be better defined. Is this effect significantly dependent on BMI or not?

5) Figure 2. The authors should add in the legend the number (n=) of pairs included in the linear regression. Moreover, the correlation between BMI and dose-corrected serum concentrations of fluvoxamine was analyzed within the whole population or just on reponders patients? Please clarify.

6) Similarly to the previous point. Was the multivariable linear regression conducted on the whole studied population or just on reponders? Moreover, was the response to treatment indicated as categorical variable? This could change the effect of BMI as predictor of serum fluvoxamine concentrations. In fact, as long as dose-corrected serum concentrations are within the therapeutic range defined by reponders patients, knowing that BMI explains for the % of variance in concentrations could be clinically irrilevant. I would rather emphatize the concept that high BMI could require an higher dosage since low serum concentrations of fluvoxamine are expected.

Minor points

Lines 397-398. This concept is reported too many times throughout the manuscript.

4. Discussion. I'm not sure whether Pharmaceutics guidelines require to divide the Discussion section in subheadings.

Comments on the Quality of English Language

The manuscript is written in a fully understanding languange and does not require deep revisions.

Author Response

Reviewer 1

Comments and Suggestions for Authors

In this manuscript by Taurines R and colleagues, the authors have analysed the relationship between serum concentrations and the dose of fluvoxamine administered to children and adolescents for treating obsessive-compulsive-disorders (OCD). Based on their multiple linear regression analysis, the authors have identified the high body mass index (BMI) as main source of variability among patients being associated with lower (dose-corrected) fluvoxamine serum concentrations. Finally, the authors have proposed a therapeutic range based on fluvoxamine serum concetrations measured in OCD responders.

In my opinion this analysis has been well conducted and conclusions seem to be adequately supported by results. However, there are some concerns that should be addressed:

1) 2.2 Measurement of fluvoxamine serum concentrations. The authors should provide more details on the bioanlytical method used. Was this method already published or validated elsewhere?

Thank you for this question. The method, developed by the TDM laboratory of Prof. Christoph Hiemke, Mainz, Germany, has already been published, validated and used for other studies. We added the source of the method to the manuscript: Härtter S, Wetzel H, Hiemke C. Automated determination of fluvoxamine in plasma by column-switching high-performance liquid chromatography. Clin Chem. 1992 Oct;38(10):2082-6.

The same method was used e.g. in Härtter S, Wetzel H, Hammes E, Torkzadeh M, Hiemke C Ninlinear pharmacokinetics of fluvoxamine and gender differences. Ther Drug Monit. 1998 Aug;20(4):446-9

2) 2.4 Data analysis. The dose used for the multiple linear regression was normalized on body weight? Moreover, the serum fluvoxamine concentration was dose-corrected?

Body weight and dose are variables in the model. As Fig. 2 shows, the correlation refers to dose-corrected fluvoxamine serum concentrations.

3) 3.3 Fluvoxamine concentration in relation to fluvoxamine dose and other covariates. The odd ratio or the parameter estimate should be added for the BMI covariate.

We added the estimate for BMI (equation y= 2.99 – 0.08 x and R2-values: R2 linear = 0.25), however this is not an odds ratio, because we performed linear and not logistic multiple regression.

4) Line 305. "Significant moderating effect" should be better defined. Is this effect significantly dependent on BMI or not?

Yes, the effect is significant (see line 306). We skipped the word „moderating” and added the equation to Fig. 2 to make this more clear.

5) Figure 2. The authors should add in the legend the number (n=) of pairs included in the linear regression. Moreover, the correlation between BMI and dose-corrected serum concentrations of fluvoxamine was analyzed within the whole population or just on reponders patients? Please clarify.

Thanks for this valuable comment: The correlation between BMI and dose-corrected serum concentrations was analyzed within all patients except for those patients with missing values for body weight and/or hight. Due to missing values for calculating the BMI in 10 patients, n= 60 pairs were included. We added this information in the legend of Fig 2.

6) Similarly to the previous point. Was the multivariable linear regression conducted on the whole studied population or just on reponders? Moreover, was the response to treatment indicated as categorical variable? This could change the effect of BMI as predictor of serum fluvoxamine concentrations. In fact, as long as dose-corrected serum concentrations are within the therapeutic range defined by reponders patients, knowing that BMI explains for the % of variance in concentrations could be clinically irrilevant. I would rather emphatize the concept that high BMI could require an higher dosage since low serum concentrations of fluvoxamine are expected.

Thanks for this valuable comment and see our answer to point 5). The regression was conducted on the whole study sample independently of therapeutic response. We emphasized the result by adding the following sentence to line 398: …” implying that patients with a high BMI might need a higher dose”

Minor points

Lines 397-398. This concept is reported too many times throughout the manuscript.

Thank you for the valuable hint. We deleted the following sentence in the first section of the discussion to avoid redundancies: ‚BMI and dose explained 25.3% of variance in concentrations, with a direct relationship between fluvoxamine dose and concentration and an inverse relationship between BMI and fluvoxamine concentration.’

  1. Discussion. I'm not sure whether Pharmaceutics guidelines require to divide the Discussion section in subheadings.

We deleted the subheadings of the discussion.

Reviewer 2 Report

Comments and Suggestions for Authors

The manuscript describes original and interesting study regarding the relationship between fluvoxamine dose and serum concentrations in children and adolescents. The Authors proposed the fluvoxamine therapeutic reference range for youths and identified BMI as a determinant of lower drug concentrations. The manuscript is well written and described in a thoughtful way. Just some justifications should be provided.

1.      Section 2.1. Setting and study population. How did the Authors estimate the sample size of the study? Is the current sample size power-based?

2.      Section 2.2. Measurement of fluvoxamine serum concentrations: The Authors declared that the HPLC-UV method was used for determination of fluvoxamine and provide some validation parameters. The procedure for samples preparation prior to the HPLC analysis should be provided.

3.      Line 283: The determined fluvoxamine concentrations were within the range of 12 – 754 ng/mL. However, the method was linear in the range of 10-730 ng/mL as declared in the line 156. Please, note that the concentration of fluvoxamine of 754 ng/mL is > ULOQ. How did you manage the samples with concentration > ULOQ? Were the samples diluted? If so, dilution factor should be evaluated.

4.      Section 3.2., lines 270-277: The description does not refer to the values presented in Table 2. It is not clear if the data were included in the manuscript. This should be clarified.

5.      Figure 2: Please add the equation expressing the influence of BMI on dose-corrected serum concentration of fluvoxamine.

Author Response

Reviewer 2

Comments and Suggestions for Authors

The manuscript describes original and interesting study regarding the relationship between fluvoxamine dose and serum concentrations in children and adolescents. The Authors proposed the fluvoxamine therapeutic reference range for youths and identified BMI as a determinant of lower drug concentrations. The manuscript is well written and described in a thoughtful way. Just some justifications should be provided.

  1. Section 2.1. Setting and study population. How did the Authors estimate the sample size of the study? Is the current sample size power-based?

Due to the observational character of the study including data mainly of a routine clinical TDM laboratory, data analyses were not preceded by a power analysis. However, the present data comprise – to the knowledge of the authors – the largest sample analysed in children and adolescents up to now.

  1. Section 2.2. Measurement of fluvoxamine serum concentrations: The Authors declared that the HPLC-UV method was used for determination of fluvoxamine and provide some validation parameters. The procedure for samples preparation prior to the HPLC analysis should be provided.

      The sample is not processed before HPLC, which presents the advantage of automatic column switching. The sample (serum) is transferred directly from the injector to the system (precolumn). We added the reference for the original bioanalytical method from the TDM laboratory of Prof. Christoph Hiemke, that was also used in our lab/in the present study: Härtter S, Wetzel H, Hiemke C. Automated determination of fluvoxamine in plasma by column-switching high-performance liquid chromatography. Clin Chem. 1992 Oct;38(10):2082-6.

  1. Line 283: The determined fluvoxamine concentrations were within the range of 12 – 754 ng/mL. However, the method was linear in the range of 10-730 ng/mL as declared in the line 156. Please, note that the concentration of fluvoxamine of 754 ng/mL is > ULOQ. How did you manage the samples with concentration > ULOQ? Were the samples diluted? If so, dilution factor should be evaluated.

Samples above a concentration of 730ng/ml were diluted. In the method section we added the following sentence: ‘Samples above a concentration of 730ng/ml were diluted with isotonic saline solution, different dilutions of the sample were measured and compared with the undiluted sample’.

  1. Section 3.2., lines 270-277: The description does not refer to the values presented in Table 2. It is not clear if the data were included in the manuscript. This should be clarified.

Thank you for your hint, that the reference to table 2 is misleading. The following data is shown in table 2 (line 270/271), and the sentence has been clarified as follows: ‘Patients of the whole transdiagnostic sample were treated with an average of 140.4 mg/d (SD 61.4; range 25-300 mg/d) fluvoxamine (Table 2)‘.

We adapted the next sentence as follows: ‚The group of patients with OCD (n=53) was treated with significantly higher daily doses (149.5±62.7mg/d) compared to all patients (n=15) with other diagnoses (113.3±50.8mg/d) (t=-2.050; p=0.044); data on the whole group of patients with OCD and the whole group of patients with other diagnoses are not additionally shown in table 2’.

  1. Figure 2: Please add the equation expressing the influence of BMI on dose-corrected serum concentration of fluvoxamine.

We added the equation and R2 values: y= 2.99 – 0.08 x; R2 linear = 0.25.

Reviewer 3 Report

Comments and Suggestions for Authors

The manuscript describes analysis of “real world” data on fluvoxamine serum concentration - dose relationship in pediatric and adolescent patients diagnosed with OCD. The manuscript presents interesting data on fluvoxamine TDM in these patients and their analysis. The manuscript is well organized and is well written, in a professional and balanced fashion. The figures and graphs are of high quality and informative. I have only small comments to the contents of this manuscript.

Minor comments:

Line 179 – please state whether 1-tail or 2-tail test was used for data analysis.

Figure 2 – please add the equation and R2 values (as shown in Fig. 1).

Author Response

Reviewer 3

Comments and Suggestions for Authors

The manuscript describes analysis of “real world” data on fluvoxamine serum concentration - dose relationship in pediatric and adolescent patients diagnosed with OCD. The manuscript presents interesting data on fluvoxamine TDM in these patients and their analysis. The manuscript is well organized and is well written, in a professional and balanced fashion. The figures and graphs are of high quality and informative. I have only small comments to the contents of this manuscript.

Minor comments:

Line 179 – please state whether 1-tail or 2-tail test was used for data analysis.

Thank you for the valuable hint. 2-tailed tests were used, which was added to the statistic section.

Figure 2 – please add the equation and R2 values (as shown in Fig. 1).

Thank you for this helpful comment. We added the equation and R2 values: y= 2.99 – 0.08 x; R2 linear = 0.25.
